# Development of an Evaluation Index for Forest Therapy Environments

**DOI:** 10.3390/ijerph21020136

**Published:** 2024-01-25

**Authors:** Jaewoo Kang, Jeongho Choi, Kyoungmin Lee

**Affiliations:** Forest Welfare Research Center, Korea Forest Welfare Institute, Yeongju-si 36043, Republic of Korea; kjw0523@fowi.or.kr (J.K.); uptake@fowi.or.kr (J.C.)

**Keywords:** forest therapy environment, evaluation indicator, forest therapy resource, forest therapy, indicator development

## Abstract

Most research on forest therapy has examined the therapeutic effects of forest activity development. There has been insufficient research identifying and evaluating the forest therapy environment. This study aimed to derive a representative forest therapy environment from each of the four evaluation sites, comprising national luxury forests; Scopus, PubMed, Medline, Web of Science, RISS, and DBpia were searched, and 13 studies evaluating forest therapy environments were analyzed and synthesized. After conducting a Conformity Evaluation, one layer of items, comprising anions with low conformity scores, was excluded, and six field measurements, phytoncide, oxygen, illuminance, UV-rays, sound, and anion, were added to increase objectivity. Finally, five forest therapy environment categories and 25 detailed items were derived. Analytic Hierarchy Process-based importance was evaluated to calculate the weight between the final evaluation items. According to the site evaluations, the categories of landscape, forest air, sunlight, sound, and anions appeared, in that order. This study is significant as it developed evaluation items and rating criteria for forest therapy environments, applied these in the field, and derived representative forest therapy environments for each location. This study developed indicators, provided basic data for establishing a therapy environment management plan, and there recommendations were made for an environment suitable for visitors and customizing forest welfare and therapy services.

## 1. Introduction

In response to the increased interest in health, modern society has been increasingly demanding the use of natural environments, such as forests, as therapy or treatment spaces to improve their quality of life [1,2]. To accommodate the increasing interest in health promotion and leisure activities, which reduce the likelihood of disease, various natural environments for trekking, hiking, and forest bathing are required [3]. In addition, health promotion activities using the natural environment are naturally increasing.

Several countries have implemented the use of natural environments to promote the health of residents. From the end of the 19th century in Germany, Father Sebastian Kneipp began to promote various therapies that used water temperature and pressure for recreational and medical purposes. Currently, there are more than 53 nationally approved Knife health resorts in Germany for disease prevention and treatment, and therapy programs for people with diseases such as chronic respiratory diseases are in operation [4]. The United Kingdom conducted a natural prescription study to provide national health services linked to medical institutions. This study revealed that it effectively reduces blood pressure, stabilizes the mind and body, and improves happiness in patients with diseases such as high blood pressure. In the United States, natural prescriptions have been made since 2013 to reduce the burden of medical costs due to chronic diseases. As of 2018, 32 US states are implementing 71 natural prescription programs [5]. Since 2005, Japan has created forest therapy roads and bases that have been utilized to promote health. A forest therapy base is a designated area with certain therapy effects that is used to treat city residents’ mental and physical conditions [6]. Additionally, Australia and Italy have conducted research on the positive effects of forests on human health [7,8].

As these studies show, most research into forest therapy has been conducted to verify the effectiveness of activity development. However, these studies are limited in that they either focus on forest activities or examine their therapy effects. Research identifying and evaluating the forest therapy environment, which is the basis of forest therapy, is insufficient. Factors such as landscape, forest air, oxygen, phytoncide, sunlight, sound, and anions are important in enabling forest therapy; nonetheless, studies of these factors are limited.

Phytoncide released from the forest is a natural volatile organic compound with a bactericidal effect produced by plants and is mainly composed of terpenes such as pinene, limonene, and campene. As a study related to phytoncide, inhalation of air with phytoncide had a psychological and physiological stability effect [9,10]. In addition, cardiovascular and stress indicators were improved in the group that inhaled phytoncide and exercised at the same time compared to the exercise-only group [11,12]. In the case of mouse experiments using phytoncide, the higher the concentration of phytoncide injected into cells within the appropriate range, the higher the physiological recovery effect such as wound recovery [13,14]. In the case of other forest healing environment studies, when appreciating the forest landscape, they became physiologically stable, such as brain activity, pulse, muscle, and blood pressure [15,16]. Sunlight improves depression, and vitamin D, produced by sunlight, aids in the absorption of calcium and phosphorus and strengthens bones [17]. Unlike cities, the sound of nature was found to have evenly distributed energy and many alpha waves were measured in the sound of waterfalls [18]. Therefore, this study aimed to develop a method for evaluating the forest therapy environment that could address this research gap.

The Korean Forest Service has evaluated the feasibility of therapy forests under the Forest Cultural Recreation Act. However, only phytoncides, anions, and altitude were considered when evaluating the therapy environment. In a related study that aimed to evaluate forest therapy environments, conduct a forest landscape resource survey, and develop a forest landscape rating, ultraviolet index, natural sound classification, and anion rating in mountain-type parks, some components and methods for evaluating landscapes, sunlgiths, sounds, anions were examined [19,20,21,22,23]. However, the forest therapy environment in that study dealt with one item and had subjective limitations such as no field measurements. As with literature related to the forest therapy environment evaluation index, there was a development of a forest therapy index and an evaluation model of therapy forests [24,25]. In these studies, when evaluating the suitability and importance of the evaluation index, it was conducted in the form of asking expert opinions without using sophisticated analysis techniques such as CVR and AHP analysis. In addition, the study was limited to systematically evaluating the comprehensive forest therapy environment because the forest therapy environment was not classified into environmental items and detailed items, and no importance was given to each. Recently, Korea introduced forest welfare facilities as part of a nationwide therapy forest creation project. Therefore, it is necessary to prepare a therapy environment management plan that can be used as the number of forest welfare facilities increases.

Additionally, this study presented indicators for analyzing and evaluating the current status of forest therapy environments systematically and objectively. Whether a facility is excellent or insufficient can be determined by evaluating the therapy environments of the major forest welfare facilities nationwide. Thus, it is possible to provide a therapy environment management plan that manages excellent environments and supports inadequate ones. Furthermore, by deriving a representative forest therapy environment for each facility, it is possible to recommend a customized therapy environment desired by visitors.

This study presented a plan to develop evaluation indicators for managing the forest therapy environment based on target sites. First, a literature review was used to derive the evaluation items of forest therapy environments, and the rating criteria for each item were analyzed. Second, the suitability and importance of the evaluation items were evaluated using a Delphi survey. Third, the validity of the forest therapy environment evaluation index was determined through field evaluation.

## 2. Materials and Methods

Evaluation indicators, that is, evaluation items and rating criteria, are useful when quantitatively evaluating the environment [26,27]. This study used a three-stage research method. First, a systematic literature survey was conducted by comprehensively collecting and analyzing existing data on the forest therapy environment evaluation items and rating criteria. Evaluation items and rating criteria were organized around the landscape, forest air, sunlight, sound, and anion presented by the Korea Forest Service.

Second, a Delphi survey was conducted to evaluate the suitability and importance of evaluation items [28,29,30,31]. The Delphi technique is frequently used in science and technology developments as well as forest policymaking [32,33,34]. Therefore, this study can use this method sufficiently to evaluate the suitability and importance of the forest therapy environment evaluation index. The content validity ratio (CVR) was used to assess the suitability of the evaluation items. The content validity (CV) method is frequently used as a criterion for determining items that should be deleted or added when developing an evaluation index [35]. Moreover, the analytic hierarchy process (AHP) technique was used to evaluate the importance of items.

Third, the validity of the forest therapy environment evaluation index was verified through field evaluation. Four national luxury forests in Korea (i.e., the Inje birch forest, Cheongoksan Ecological Management Forest, Sinbul Mountain silvergrass forest, and Yeongju Masil Therapy Forest) were evaluated using climate. The score was calculated by reflecting the importance of the field evaluation value of the forest therapy environment. Based on the results, a management plan for the therapy environment of luxury forests was derived.

The study period was from April 2021 to September 2023 (Figure 1).

### 2.1. Literature Review

#### 2.1.1. Literature Selection Criteria

This study conducted a systematic literature review by comprehensively collecting and analyzing existing data on evaluation items and rating criteria for forest therapy environments. The literature selection process was conducted in accordance with the PRISMA (Preferred Reporting Items for Systematic Review and Meta-analysis) [36] guidelines. This included papers and dissertations published in academic journals, prioritized academic papers when academic papers and dissertations overlapped, and excluded overlapping research papers.

#### 2.1.2. Data Search and Selection Process

The literature search focused on papers published both in Korea and globally between 1 January 2020, when forest therapy began to be studied in earnest, and 31 May 2022. The languages were limited to English and Korean, and the search was conducted for two months, from 5 April to 4 June 2022. Scopus, PubMed, MEDLINE (EBSCO), Web of Science, RISS, and DBpia databases were reviewed. Keywords were centered on the representative therapy resources suggested by the Korea Forest Service. We searched for the following keywords: “forest therapy resource,” OR “forest healing resource,” OR “forest landscape,” OR “forest air,” OR “phytoncide,” OR “oxygen,” OR “sunlight,” OR “sound,” OR “anion.”

The selection criteria were as follows: (1) forest therapy resource items presented by the Korea Forest Service and (2) literature that specified the basis for the concept and rating standard. Review papers, cases or qualitative studies, and literature with no results were excluded. Duplicate documents were excluded using EndNote X9, a bibliographic management program. Seven forest majors who had studied in the forest field for more than five years each reviewed and confirmed the title and abstract of each paper to ensure it examined forest therapy resources. Subsequently, the full text was checked to determine whether the selection and exclusion criteria were met. Data on author, year, and study design were extracted from all studies that met the qualifying criteria. The 16 selected studies were classified according to the forest therapy environment.

### 2.2. Delphi Survey

The Delphi survey evaluated the suitability of the forest therapy environment evaluation items selected during the literature survey. A conformity evaluation using the CVR was conducted, and the questions were revised and supplemented according to expert opinions. The AHP-based importance was assessed for the final evaluation items, and the weights of each item were calculated. In this study, 45 people with more than one year of experience, such as professors, researchers, and instructors in the field of forest therapy or forest welfare or forestry, were selected.

The CVR uses a technique proposed by Lawshe [37] to evaluate the suitability of the evaluation item. The CVR objectively verifies the CV based on the percentage of expert panel consensus on the appropriateness of each evaluation item. Subsequently, the CVR formula (Figure 2) was derived. A 7-point Likert scale (where 1 = very unsuitable, and 7 = very suitable) was utilized by the expert panel to rate the items. In Figure 2, “$n_{e}$” represents the number of panels that responded to the fit (5–7 points) of the evaluation item, and “N” represents the total number of panels that responded. The CVR values ranged from +1 to −1, with +(plus) indicating that at least half of the panel’s responses indicated that the item was appropriate. According to Ayre and Scally [38], the minimum required value was set to 0.30 when there were 40 respondents.

AHP-based importance was assessed for the final evaluation items to calculate the weight of each item. The AHP, developed by Saaty [39], is a technique that determines priorities by stratifying multiple attributes and identifying the importance of each attribute. The AHP derives a consistency index in the process of integrating pairwise comparison results to check the logical consistency of decision-makers. When the consistency index (CI) exceeds 0.1, the decision is reviewed to ensure the logic and rationality of the decision maker. The AHP is applied in various fields, such as policymaking, decision-making, and follow-up project evaluation data (Figure 3) [40,41,42,43]. In Figure 3, “A” is the property of a pairwise comparison matrix, and “aij” is the property of a pairwise comparison matrix “A”. Furthermore, “n” is the number of elements to be compared within a layer, and “ωi” means the relative importance of “n” elements. In this study, if the CI or consistency ratio (CR) of the panel responses was 0.1 or more, the item was excluded from the analysis.

### 2.3. Field Evaluation

The sites were assessed to validate the final forest-therapy environment evaluation index. As for the evaluation sites, 40 national luxury forests and the top 100 mountains nationwide, which are major forests in Korea designated by the Korea Forest Service, were reviewed. As a result, four national luxury forests located in different climates across Korea (See Figure 4) with clearly designated areas and scopes were selected among places; these are well-preserved forests that have excellent ecological value.

The Inje birch forest is located in Inje-gun, Gangwon-do. It has a 6 ha scale with a tree diameter of 14 cm and a height of 10 m with 5500 birch trees. The landscapes are full of white bark trees.

The Cheongoksan Ecological Management Forest is located in Bonghwa-gun, Gyeongsangbuk-do, in an area over 1000 m above sea level. This forest is designated as a “forest genetic resource protection zone” due to the various tree species, such as oak and ash trees, growing around the Geumgang pine forest.

The Sinbul Mountain silvergrass forest is located in Ulju-gun, Ulsan Metropolitan City, at the top of Sinbul Mountain, about 1159 m above sea level. There are large rock formations in the east and a 3 ha silvergrass plain along a 4 km ridge on the mountain.

The Yeongju Masil Therapy Forest is located in Yeongju, Gyeongsangbuk-do, and has several different tree species, such as pine trees and Japanese larch trees. It is a forest where one can appreciate nature along the deck road and walk through low-slope dense forest.

The site was evaluated according to the evaluation index of the forest therapy environment derived from the Delphi survey. First, three forest therapy experts analyzed the type of tree, density, and altitude of the research site in advance using ArcGIS Desktop 10.8.1, a geographic information system software, and satellite maps before conducting the field evaluation. Subsequently, three experts visited the national luxury forests and evaluated the forest therapy environment index both at each site and through discussions. Comprehensive scores for each forest therapy environment evaluation index were derived.

To obtain the field measurements of the forest therapy environment equipment, the phytoncide concentration measurement and analysis method were applied to evaluate the feasibility of the therapy forest survey site, as reported by the National Forest Research Institute [44]. For the phytoncide field measurements, an arithmetic average was performed on the values measured for 1 h at 08:00, 12:00, and 17:00, respectively.

Phytoncide adsorption tubes (Tenax TA, KNR, Namyangju, Republic of Korea) were conditioned at 310 °C for 2 h, the day prior to obtaining the measurement. In the field installation, three mini pumps (MP-Σ30KNII, Sibata, 2016, Seoul, Republic of Korea) were installed horizontally at a height of 1.5 m from the ground. After connecting the adsorption tube to the mini-pump, 9 L of air was collected at 08:00, 12:00, and 17:00 over 60 min at a flow rate of 150 mL/min. A gas chromatography/mass spectrometer (GC/MS; Shimadzu, Seoul, Republic of Korea), equipped with a thermal desorption device (TD-20; Shimadzu, Seoul, Republic of Korea), was used to analyze the adsorption tube after recovering it. The measurements were averaged at each time interval.

Oxygen field measurements were averaged using an oxygen meter (H41-H5, HiMAX Tech Co., Seoul, Republic of Korea) for 1 min at 08:00, 12:00, and 17:00. Sunlight was averaged for 30 min/h using an illuminance meter (Tenmars TM-203, AZPLUS, Seoul, Republic of Korea) and an ultraviolet meter (Delta OHM HD2302.0, AZPLUS, Seoul, Republic of Korea). A sound measurement device (H5 handy recorder, MIDI AND SOUND, Seoul, Republic of Korea) was used to apply Gim et al.’s [22] research method. According to Gim et al. [22], sounds were categorized into three categories: biological (animals, birds, and insects), inanimate (wind, rain, thunder, and water), and artificial (people, cars, music, airplanes, cleaning, and construction). From 08:00 to 17:00, one point was given if a sound was heard at least once for each sound category. Based on the sound classification, the following score ranges were determined: from 0 to 30 points for biological sounds, from 0 to 40 points for inanimate sounds, and from 0 to 60 points for artificial sounds. The sound field measurement value was calculated using the following formula: biological sound + inanimate sound-artificial sound. Anions were arithmetically averaged for 15 min per period using an anion meter (COM-3600F; NICO, Seoul, Republic of Korea) (Table 1).

## 3. Results

### 3.1. Research Results

A total of 792,230 studies were found through database searches (Scopus, N = 178,449; Pubmed, N = 195,611; MEDLINE, N = 13,073; Web of Science, N = 56,829; RISS, N = 231,619; and DBpia, N = 116,649). After deduplication, 400,453 remained. Of these, 400,327 did not match the research topic and were excluded based on the criteria for selecting and excluding data focusing on titles and abstracts. A total of 126 papers were selected, and their full texts were confirmed. Consequently, 113 studies were excluded, including those that could not be fully confirmed, those that did not meet the evaluation criteria, those not written in English or Korean, and irrelevant formal documents (Figure 5).

The publication years of the final 13 selected studies were as follows: one (8%) from 2000 to 2005, two (15%) from 2006 to 2010, three (23%) from 2011 to 2015, six (46%) from 2016 to 2020, and one (8%) after 2021. The forest therapy environment index comprised the following: five landscapes (38%), two forest air (15%), two sunlight (15%), two sound (15%), and two anions (15%).

To develop the forest therapy environment evaluation items, we divided them into five forest therapy environments and 20 items. The evaluation criteria were rated on a scale ranging from 1 to 5 points (1 = Low, 2 = Low-Intermediate, 3 = Intermediate, 4 = Upper-Intermediate, 5 = Advanced). Owing to the lack of prior research in the form of published literature, we also considered relevant laws, such as research and development reports, and detailed criteria for feasibility evaluations, such as natural recreation forests [45], to derive the criteria in this study (Table 2).

The landscape comprised five detailed items: viewpoint, specificity (colonies or water system), disturbance factor, history and culture, and wood grade. The higher the number of viewpoints, the higher the score, whereby five points were given when there were more than seven viewpoints and one point was given when there were no viewpoints [46,47]. For specificity (colonies or aquatic environments), the more unusual the plant colonies or aquatic environments, the higher the score; five points were given when there were more than four plant colonies or aquatic environments, and one point was given when there were none [23,46]. For the disturbance factor, five points were given when there was no disturbance factor, and one point was given when there were four or more damaged areas and artificial structures [19]. For history and culture, five points were given when there were five or more historical and cultural properties with forests, and one point was given when there were one or two historical and cultural properties with forests [23]. The wood grade was divided into five points for large tree diameter and one point for small tree diameter [24].

For forest air, three detailed items were selected: average altitude, type of tree (coniferous, broadleaf, and mixed forests), and designated forest area (excluding bare lands). The average altitude was given a high point value because the oxygen concentration was high when the altitude was low; five points were given when the altitude was less than 100 m, and one point was given when the altitude exceeded 700 m. The type of tree evaluated the amount of carbon fixation, with higher amounts indicating higher scores. Broadleaf forests have relatively high carbon fixation and were given five points; coniferous forests have relatively low carbon fixation and were given one point [48]. The designated forest area (excluding bare lands) was given a high number of points because the amount of oxygen generated was high when the area was large; five points were given when the area exceeded 2000 ha, and one point was given when the area was less than 200 ha.

For sunlight, three items were selected: type of tree, crown density, and cloudiness. The type of tree evaluated the score highly according to the large amount of sunlight. To assess the type of tree, five points were given to broadleaf forests, which reflect a large amount of sunlight, and one point was given to coniferous forests, which reflect less sunlight. The crown density was evaluated according to the sunny environment; five points were given for low density (40% or less), and one point was given for high density (70% or more). For the cloudiness, five points were given when it was clear, and one point was given when it was cloudy [49,50].

For sound items, four factors were selected: traffic (roads or trains), living area, water system, and bird diversity. For traffic, a higher score was given when there was less traffic; five points were given if there were no roads or trains, and one point was given if there were both. For the living area, five points were given if the forest was not around living areas, and one point was given if the forest was near metropolitan cities. For the water system, five points were given when there was an audible water system in the forest, and one point was given when there was no audible water system. For bird diversity, five points were given when there was a high diversity of potential bird habitats, such as in mixed forests, and one point was given when there was a low diversity of structures, such as in coniferous forests [51].

For the anion, five detailed items were selected: water systems, type of tree, wood grade, stratification, and crown density. For all items, a higher score was given when the number of anions was high. For the water system, five points were given when there was a large amount of anions due to the valleys and small rivers in the forest, and one point was given when there were no anions in the forest’s valleys and small rivers. For the type of tree, five points were given to broadleaf forests, which have relatively high anion generation, and one point was given to coniferous forests, which have relatively low anion generation. For the wood grade, five points were given for large-diameter trees, which have high anion generation, and one point was given for small-diameter trees, which have a relatively small anion generation. For stratification, five points were given for a single layer, and one point was given for bare lands, etc. For the crown density, five points were given for medium density (40% to 70%), and one point was given for low density (40% or less) [21].

The literature review showed that wood grade, type of tree, crown density, and water system were the items that affected the two forest therapy environments. It was determined that this factor was important because it comprised items from each forest therapy environment. The wood grade consisted of landscape and anion, the type of tree and crown density consisted of sunlight and anion, and the water system consisted of detailed items of sound and anion.

### 3.2. Delphi Survey Results

A total of 45 experts participated in the suitability evaluation and 41 in the importance evaluation. Their detailed characteristics are presented in Table 3.

The CVR-based suitability of the forest therapy environment was also evaluated. The evaluation items are 26 detailed items, including viewpoints in five forest therapy environments, selected as a result of the literature review. Consequently, 19 detailed items, such as viewpoints, were higher than the minimum required value of 0.3. Only the detailed parameters of the anion were 0.2, which was lower than the minimum required value of 0.3; therefore, these were excluded from the analysis (Table 4).

When evaluating suitability, the objectivity of each item was increased by adding equipment field measurements. Therefore, the six equipment field measurements, discussed in Section 2.3, were added as detailed items.

The two measurements of phytoncide and oxygen were added to the “forest air” category. The higher the phytoncide and oxygen levels, which have beneficial effects on the human body, the higher the score [52]. For phytoncide measurement, five points were given when the measurement value was 200 ppt/day or more, and one point was given when the measurement value was less than 70 ppt/day. For the oxygen measurement, five points were given when the measurement value was 20.7–23.5%, and one point was given when the measurement value was less than 20.3%.

The two measurements of illuminance and UV-rays were added to the “sunlight” category. The higher the amount of forest sunlight reflected through the crown, the higher the score was. For the illuminance measurement, five points were given when the measurement value was above 6793 lx, and one point was given when the value was less than 1636 lx. For the UV-ray measurement, five points were given when the measurement value was 6.1 *µ*W/cm^2^ or higher, and one point when the value was less than 0.9 *µ*W/cm^2^.

The sound measurement value was added to the “sound” category. The higher the sound measurement value (biological sound + inanimate sound-artificial sound), the higher the score [22]. For the sound measurement, five points were given when the measurement value was 19 points or higher and one point when it was less than 10 points.

The anion measurement value was added to the “anion” category. The higher the anion concentration, which has a beneficial effect on improving the body’s immunity, the higher the score. For the anion measurement, five points were given when the measurement value was more than 2000 mg/kg/day, and one point was given when the value was less than 150/day/day [53].

The field measurements for forest air (phytoncide, oxygen), sunlight (illuminance, UV-rays), and sound measurements were rated on a scale from 1 to 5 points. The anion measurement adhered to the criteria for evaluating the feasibility of therapy forests under the Forest Cultural Recreation Act of the Korea Forest Service (Table 5).

Finally, five forest therapy environments and 25 detailed items were derived.

Subsequently, 41 people evaluated the importance of applying the AHP technique to the final forest therapy environment. If the CI or CR was lower than 0.1, it was excluded. The landscape category (0.35) was the most important of the five forest therapy environment categories, followed by forest air (0.29), sunlight (0.17), sound (0.14), and anion (0.05). (Table 6).

The importance of the detailed items for each of the forest therapy environment categories was evaluated. The results are shown in Table 7.

### 3.3. Results of the Field Evaluation

To verify the validity of the final forest therapy environment evaluation index items, an on-site evaluation of four national luxury forests was conducted. The score was calculated by reflecting the importance of the seasonal (spring, summer, fall) field evaluation and the arithmetic mean. The score range for each forest therapy environment was 7.00 to 35.00 points for the landscape category, 5.80 to 29.00 points for forest air, 3.40 to 17.00 points for sunlight, 2.80 to 14.00 points for sound, and 1.00 to 5.00 points for anion. The overall score ranged from 20.00 to 100.00.

As a result of the calculation, the landscape score of the Sinbul Mountain silvergrass forest was the highest, at 24.50 points. In terms of forest air, sound, and anions, the Cheongoksan Ecological Management Forest had the highest values at 18.13, 12.97, and 3.46 points, respectively. The Inje birch forest had the highest sunlight score of 9.84 points. The Cheongoksan Ecological Management Forest had the highest total score of 64.22 points, followed by the Sinbul mountain silvergrass forest with 62.73 points, the Yeongju Masil therapy forest with 56.70 points, and the Inje birch forest with 55.99 points (See Table 8).

The Inje Birch Forest, which is a broadleaf forest, had the highest sunlight score. In terms of the Cheongoksan Ecological Management Forest, which has a dense forest environment and is over 1000 m above sea level, its widest designated area (1075 ha) was found to have high forest air, sound, and anion scores. The Sinbul Mountain silvergrass forest, which has the largest number of viewpoints, with 3 ha of silver-grass plains spread along the main ridge, showed high landscape scores (Table 9).

## 4. Discussion

This study developed indicators that can objectively and systematically evaluate forest therapy environments, as well as provide basic data for establishing a therapy environment management plan. Literature was investigated to develop the evaluation indicators for forest therapy environments, and the suitability and importance of these indicators were evaluated using a Delphi survey. Subsequently, an onsite evaluation was conducted to determine the validity of the evaluation index items.

Based on a systematic literature review, five forest therapy environment categories and 20 detailed items were selected. As a result of the conformity evaluation, conducted as part of the Delphi survey, detailed items on the layer of anions whose conformity score was lower than the minimum required value of 0.3 were excluded. Six equipment field measurement values were added to increase the objectivity of each item. Finally, five items related to the forest therapy environment were derived.

The importance of applying the AHP technique in the final evaluation of the forest therapy environment categories was assessed. This assessment revealed that landscape (0.35) was the most important category of the five forest therapy environment categories, followed by forest air (0.29), sunlight (0.17), sound (0.14), and anion (0.05).

In terms of the items with the highest importance for each of the five forest therapy environments, landscape was the viewpoint (0.35), forest air was the type of tree (0.30), sunlight was crown density (0.29), sound was bird diversity (0.33), and anion was anion measurement (0.35). The importance evaluation of the items is helpful in identifying and managing items with the highest importance first.

The sites were assessed to verify the validity of the forest therapy environment evaluation index. The Inje birch forest had the highest sunlight score. The Cheongoksan Ecological Management Forest had the highest scores for forest air, sound, and anions. The Sinbul Mountain silvergrass forest, which has 3 ha of silvergrass plains and seven viewpoints on the main ridge, had the highest landscape score. These results are judged to have been derived in accordance with the characteristics of the major type of tree, designated areas (ha), water systems, and viewpoints for each place.

Research on the development of quantitative evaluation indicators has the advantage of being concise and easy to apply. These studies objectively evaluated forest therapy environments and are believed to serve as a basis for analyzing and evaluating the current status of the forest therapy environment. Therefore, this study derived comprehensive scores for each therapy environment by developing forest therapy environment indicators and conducting field evaluations. The evaluation results are expected to serve as basic data for establishing a management plan for forest therapy environments in each location.

A forest therapy environment with high scores at each location is considered as being representative. Forest therapy environments with low scores could be used as a reference to compensate for an insufficient environment. If a representative forest therapy environment is derived for each forest welfare facility, an environment suitable for visitors can be recommended. Thereafter, this data can be used to provide customized forest welfare and forest therapy services for the entire nation.

The results of this study can be used in academia, administration, and industry. Academia has shown that if the development of forest therapy activities or the verification of their effects is the focus [54,55,56,57,58,59,60], it is necessary to examine and study the forest therapy environment more closely in the future. The administrative community should institutionalize scientific design and management measures to ensure that the therapy environment forests, designated as national forests, are properly maintained according to each demand, and not just to create forest welfare facilities [61,62]. Industry proposes the creation of an environment or the development of new products that can use such forest therapy environments, even for consumers who cannot visit forests [63,64].

This study aimed to develop standardized evaluation indicators for the entire forest therapy environment. Therefore, there is a limit to evaluating the effects of forest therapy environments on groups of people, such as patients with diseases. Therefore, further research is needed to develop evaluation indicators of the effects of forest therapy environments for patients with diseases and to recommend forest therapy environments to special subjects. In addition, although some previous studies have discussed the effects of anions and oxygen in forests [21,50,53,65,66], further research is necessary to re-examine their impact on the human body and to supplement the forest therapy environment evaluation index. During the Delphi survey, when evaluating the importance of applying the AHP technique, if the CI or the CR was 0.1, it was immediately excluded. Therefore, the number of people included in the AHP analysis decreased. In future Delphi surveys, if the CI or CR is 0.1 or higher, it will be necessary to induce re-questioning to secure more expert data. In subsequent studies, additional statistical verification, including Cronbach’s alpha, collinearity analysis, and factor analysis, is required to ensure the validity of the developed evaluation index. Securing data evaluated on-site with more experts is essential for this purpose.

Once the reliability of the forest therapy environment evaluation index is established, the nationwide assessment of forest therapy environments becomes feasible. This, in turn, enables the recommendation of superior forests and those with specialized environments to visitors. Moreover, by supplementing the certification criteria for therapy forests in each country, including the evaluation of the feasibility of therapy forests, there is a significant opportunity to contribute to the enhancement of health and quality of life. Certifying forests with proper therapy environments and encouraging people to visit them can play a crucial role in achieving this goal.

## 5. Conclusions

This study developed indicators to objectively and systematically evaluate forest therapy environments. An evaluation index was derived from a systematic literature review. The suitability and importance of the items were evaluated using a Delphi survey, and five forest therapy environments and 25 detailed items were derived. The national luxury forests were evaluated onsite as an evaluation index for the developed forest therapy environment. The Inje birch forest had high scores for sunlight and Cheongoksan Ecological Management Forest had high scores for forest air, sound, and anions, as well as the Sinbul Mountain silvergrass forest had high scores for landscape. It was possible to derive a management plan for the forest therapy environment at each location by referring to the evaluation results. Additionally, it can be used as a basis for providing customized forest welfare services by recommending a derived representative forest therapy environment for each target.

## Figures and Tables

**Figure 1 ijerph-21-00136-f001:**
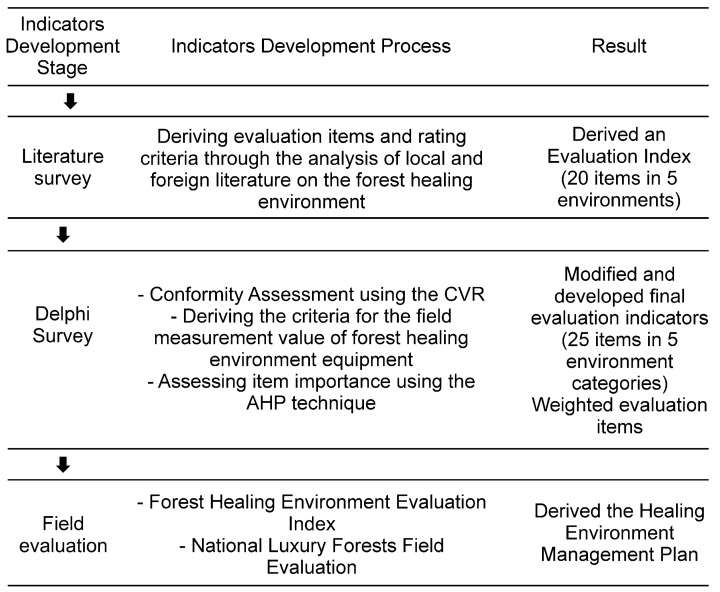
Development of the forest therapy environment evaluation index and field evaluation process.

**Figure 2 ijerph-21-00136-f002:**
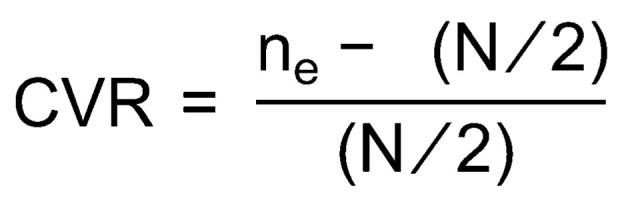
Content validity ratio (CVR) formula.

**Figure 3 ijerph-21-00136-f003:**
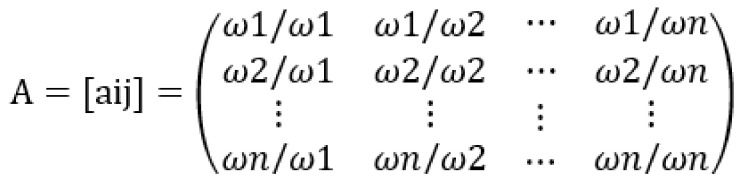
Pairwise matrix comparison.

**Figure 4 ijerph-21-00136-f004:**
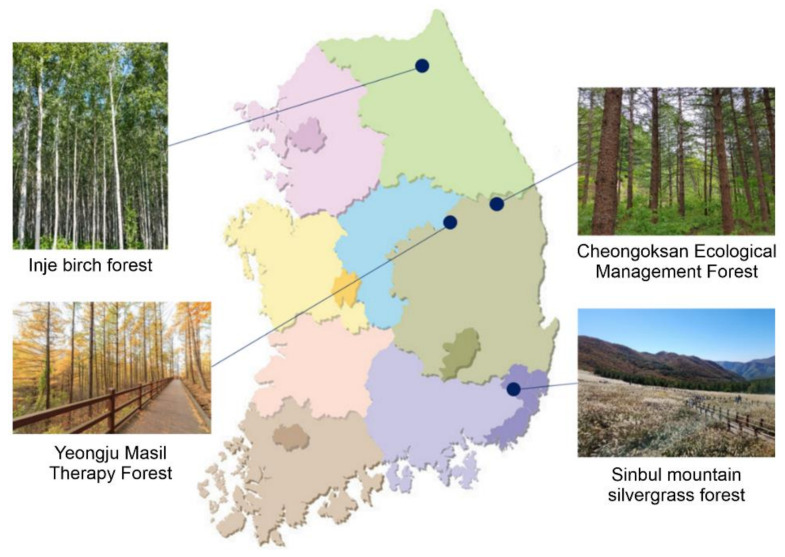
Study sites (four national luxury forests).

**Figure 5 ijerph-21-00136-f005:**
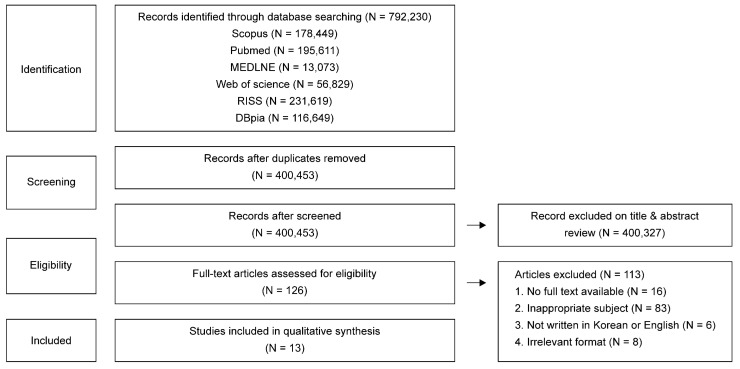
PRISMA flow chart.

**Table 1 ijerph-21-00136-t001:** Measurement time, equipment, and units by forest therapy resource.

Sortation		Measurement Time	Measurement Equipment	Unit of Measurement	Measurement Data
Phytoncide		Measured three times a day (8:00, 12:00, 17:00), averaged for 60 min at a flow rate of 150 mL/min(9 L) at each time interval	Collection: MP- ∑30KNII Analysis: TD-20, GC/MS	pptv	Phytoncide measurement
Oxygen		Measured three times a day (8:00, 12:00, 17:00), averaged for 1 min at each time interval	H41-H5	%	Oxygen measurements
Sunlight	Illuminance	Measured three times a day (8:00, 12:00, 17:00), averaged for 30 min at each time interval	Tenmars TM-203	lx	Illuminance measurement
	UV rays		Delta OHM HD2302.0	*µ*W/cm^2^	UV measurement
Sound		Recorded 10 times a day (from 8:00 to 18:00), added for 1 min at each time interval	H5 handy recorder	Score	Biological, inanimate, and artificial sound scores; sound measurement
Anion		Measured three times a day (8:00, 12:00, 17:00), averaged for 15 min at each time interval	COM-3600F	Number/cm^3^	Anion measurement

**Table 2 ijerph-21-00136-t002:** Evaluation index of the forest therapy environment.

Evaluation Criteria						
Category	Detailed Items	Advanced (5 Points)	Upper-Intermediate (4 Points)	Intermediate (3 Points)	Low-Intermediate (2 points)	Low (1 Point)
Landscape (5)	Viewpoint [46,47]	≥7 locations	5–6 locations	3–4 locations	1–2 locations	None
	Specificity	≥4 plant colonies or water system	three plant colonies or water system	two plant colonies or water system	one plant colony or water system	None
	Disturbance factor [19]	No disturbance	one damaged area or artificial structure)	two damaged areas or artificial structures	three damaged areas or artificial structures	≥4 damaged areas or artificial structures
	History and culture	≥5 historical and cultural properties with forests	-	3–4 historical and cultural properties with forests	-	1–2 historical and cultural properties with forests
	Wood grade [24]	Large tree diameter (≥30 cm)	-	Medium tree diameter (≥18 cm to <30 cm)	-	Small tree diameter (≥6 cm to <18 cm)
Forest air (3)	Average altitude	Under 100 m	101–200 m	201–400 m	401–700 m	Over 700 m
	Type of tree [48]	Broadleaf forest	-	Mixed forest	-	Coniferous forest
	Designated forest area (excluding bare lands)	>2000 ha	1001–2000 ha	501–1000 ha	201–500 ha	200 ha ≥
Sunlight (3)	Type of tree [49,50]	Broadleaf forest	-	Mixed forest	-	Coniferous forest
	Crown density	Small (40% ≥)	-	Medium (40% to 70%)	-	Large (≥70%)
	Cloudiness	Clear	A little cloudy	Cloudy	A lot of cloud	Cloudiness
Sound (4)	Traffic (roads or trains)	Neither roads nor trains	-	Either roads or trains	-	Both roads and trains
	Living area	Not around the living area	-	Near the city	-	Near metropolitan cities
	Water system	Water system in the forest	-	A nearby water system exists	-	No water system in the forest
	Bird diversity [51]	Mixed forest	-	Broad-leaved forest	-	Coniferous forest
Anion (5)	Water system	Including valleys and small rivers in the forest	-	Including small streams in the forest	-	None
	Type of tree [21]	Broadleaf forest	-	Mixed forest	-	Coniferous forest
	Wood grade	Large tree diameter (≥30 cm)	-	Medium tree diameter (≥18 cm to <30 cm)	-	Small tree diameter (≥6 cm to <18 cm)
	Stratification	Single layer	-	Multi-layer	-	Bare lands, etc.
	Crown density	Medium (40% to 70%)	-	Large (≥ 70%)	-	Small (40% ≥)

**Table 3 ijerph-21-00136-t003:** Characteristics of the experts who participated in the Delphi survey.

Related Field	Suitability Evaluation ^1^	Importance Evaluation ^2^	Number of Years in the Career	Suitability Evaluation ^1^	Importance Evaluation ^2^
Forest therapy	35	31	1–5 years	29	26
Forest welfare	7	7	6–10 years	5	4
Forestry student	3	3	11 years or more	11	11

^1^ Suitability evaluation N = 45, ^2^ Importance evaluation N = 41 (unit: number).

**Table 4 ijerph-21-00136-t004:** CVR-based suitability of the forest therapy environment results.

Category	Detailed Items	Average Point
Landscape (5)	Viewpoint	0.96
	Specificity (colonies or water system)	0.82
	Disturbance factor	0.60
	History and culture	0.47
	Wood grade	0.56
Forest air (5)	Average altitude	0.60
	Type of tree	0.87
	Designated forest area (excluding bare lands)	0.56
	(add) Oxygen measurement	0.78
	(add) Phytoncide measurement	0.91
Sunlight (5)	Type of tree	0.64
	Crown density	0.73
	Cloudiness	0.47
	(add) Illuminance measurement	0.69
	(add) Ultraviolet (UV)-ray measurement	0.38
Sound (5)	Traffic (roads or trains)	0.56
	Living area	0.42
	Water system	0.96
	Bird diversity	0.64
	(add) Sound measurement	0.69
Anion (5)	Water system	0.82
	Type of tree	0.47
	Wood grade	0.51
	(delete) stratification	0.20
	Crown density	0.64
	(add) Anion measurement	0.64

**Table 5 ijerph-21-00136-t005:** The final version of the forest therapy environment evaluation index.

Evaluation Criteria						
Category	Detailed Items	Advanced (Five Points)	Upper–Intermediate (Four Points)	Intermediate (Three Points)	Low–Intermediate (Two Points)	Low (One Point)
Landscape (5)	Viewpoint	≥7 locations	5–6 locations	3–4 locations	1–2 locations	None
	Specificity	≥4 plant colonies or water system	3 plant colonies or water system	2 plant colonies or water system	1 plant colony or water system	None
	Disturbance factor	No disturbance	1 damaged area or artificial structure	2 damaged areas or artificial structures	3 damaged areas or artificial structures	≥4 damaged areas or artificial structures
	History and culture	≥5 historical and cultural properties with forests	-	3–4 historical and cultural properties with forests	-	1–2 historical and cultural properties with forests
	Wood grade	Large tree diameter (over 30 cm)	-	Medium tree diameter (≥18 cm to <30 cm)	-	Small tree diameter (≥6 cm to <18 cm)
Forest air (5)	Average altitude	Under 100 m	101 m–200 m	201 m–400 m	401 m–700 m	Over 700 m
	Designated forest area (Excluding bare lands)	>2000 ha	1001 ha–2000 ha	501 ha–1000 ha	201 ha–500 ha	200 ha ≥
	Type of tree	Broadleaf forest	-	Mixed forest	-	Coniferous forest
	Phytoncide measurement	≥200 ppt/day	Less than 200 ppt/day	Less than 170 ppt/day	Less than 130 ppt/day	70 ppt/day >
	Oxygen measurement	20.7% to 23.5%	20.6%	20.5%	20.4%	20.3% ≥
Sunlight (5)	Type of tree	Broadleaf forest	-	Mixed forest	-	Coniferous forest
	Crown density	Small (40% ≥)	-	Medium (40% to 70%)	-	Large (≥70%)
	Cloudiness	Clear	A little cloudy	Cloudy	A lot of cloud	Cloudiness
	Illuminance measurement	≥6793 lx	Less than 6793 lx	Less than 5074 lx	Less than 3355 lx	Less than 1636 lx
	UV-rays measurement	≥6.1 *μ*W/cm^2^	Less than 6.1 *μ*W/cm^2^	Less than 4.4 *μ*W/cm^2^	Less than 2.6 *μ*W/cm^2^	0.9 *μ*W/cm^2^ >
Sound (5)	Traffic (roads or trains)	Neither roads nor trains	-	Either roads or trains	-	Both roads and trains
	Living area	Not around living area	-	Near the city	-	Near metropolitan cities
	Water system	Water system in the forest	-	A nearby water system exists	-	No water system in the forest
	Bird diversity	Mixed forest	-	Broad-leaved forest	-	Coniferous forest
	Sound measurement	≥19 points	Less than 19 points	Less than 16 points	Less than 13 points	10 points ≥
Anion (5)	Water system	Including valleys and small rivers in the forest	-	Including small streams in the forest	-	None
	Type of tree	Broadleaf forest	-	Mixed forest	-	Coniferous forest
	Wood grade	Large tree diameter (≥30 cm)	-	Medium tree diameter (≥18 cm to <30 cm)	-	Small tree diameter (≥6 cm < 18 cm)
	Crown density	Medium (40% to 70%)	-	Large(≥70%)	-	Small (≥40%)
	Anion measurement	≥2000/cm^3^/days	Less than 2000/cm^3^/days	Less than 1000/cm^3^/days	Less than 700cm^3^/days	150/cm^3^/days >

**Table 6 ijerph-21-00136-t006:** Forest therapy environment AHP analysis results (N = 12).

Forest Therapy Environment	Landscape	Forest Air	Sunlight	Sound	Anion
Importance	0.35	0.29	0.17	0.14	0.05

**Table 7 ijerph-21-00136-t007:** The results of AHP analysis of evaluation items by forest therapy resources.

Ranking	1	2	3	4	5
Landscape (N = 8)	Viewpoint (0.35)	Specificity (0.24)	Disturbance factor (0.21)	History and culture (0.10)	Wooden grade (0.10)
Forest air (N = 14)	Type of tree (0.30)	Phytoncide measurement (0.25)	Oxygen measurement (0.20)	Average altitude (0.13)	Designated forest area (0.12)
Sunlight (N = 16)	Crown density (0.29)	Type of tree (0.27)	Illuminance measurement (0.16)	UV-ray measurement (0.16)	Cloudiness (0.12)
Sound (N = 10)	Bird diversity (0.33)	Water system (0.29)	Sound measurement (0.22)	Living area (0.09)	Traffic (0.07)
Anion (N = 15)	Anion measurement (0.35)	Water system (0.29)	Wooden grade (0.15)	Crown density (0.13)	Type of tree (0.08)

**Table 8 ijerph-21-00136-t008:** Field evaluation results (four national luxury forests).

Study Site	Inje Birch Forest	Cheongoksan Ecological Management Forest	Sinbul Mountain Silver-Grass Forest	Yeongju Masil Therapy Forest
Landscape	17.64	20.51	24.50	19.74
Viewpoint	2.45	4.90	12.25	4.90
Specificity	5.04	8.40	5.04	6.16
Disturbance factor	7.35	4.41	4.41	5.88
History and culture	0.70	0.70	0.70	0.70
Wood grade	2.10	2.10	2.10	2.10
Forest air	15.66	18.13	15.51	16.32
Average altitude	0.75	0.75	0.75	1.51
Type of tree	8.70	5.22	5.22	5.22
Designated forest area	0.70	2.78	2.09	0.70
Phytoncide measurement	2.42	4.35	4.35	5.80
Oxygen measurement	3.09	5.03	3.09	3.09
Sunlight	9.84	9.13	7.64	6.78
Type of tree	4.59	2.75	2.75	2.75
Crown density	0.99	0.99	0.99	0.99
Cloudiness	1.36	1.22	1.36	1.22
Illuminance measurement	1.27	2.72	1.45	0.91
UV-ray measurement	1.63	1.45	1.09	0.91
Sound	9.69	12.97	11.74	11.29
Traffic	0.98	0.98	0.98	0.59
Living area	1.26	1.26	1.26	1.26
Water system	4.06	4.06	4.06	2.98
Bird diversity	2.77	4.62	4.62	4.62
Sound measurement	0.62	2.05	0.82	1.85
Anion	3.16	3.46	3.35	2.57
Water system	0.87	1.45	1.45	0.68
Type of tree	0.40	0.24	0.24	0.24
Wood grade	0.45	0.45	0.45	0.45
Stratification	0.39	0.39	0.39	0.39
Anion measurement	1.05	0.93	0.82	0.82
Total score	55.99	64.22	62.73	56.70

**Table 9 ijerph-21-00136-t009:** Characteristics of the four national luxury forests and the representative forest therapy environment.

Study Site	Characteristics				Specialized Forest Therapy Environment
	Major Type of Tree	Designated Forest Area (ha)	Water System	Viewpoint	
Inje birch forest	Birch forest	7	A river in the forest	None	Sunlight
Cheongoksan Ecological Management Forest	Mixed forest	1075	A valley or river in the forest	Two locations	Forest air, sound, anion
Sinbul Mountain silvergrass forest	Silver-grass plain	754	A valley or river in the forest	Seven locations	Landscape
Yeongju Masil therapy forest	Mixed forest	20	Rivers in the forest (summer, fall)	Two locations	-

## Data Availability

Data are contained within the article.

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
