# Peer review of "Development of an Evaluation Index for Forest Therapy Environments"

_ijerph, 2024, doi:10.3390/ijerph21020136_

Round 1
Reviewer 1 Report
Comments and Suggestions for Authors
Thank you for allowing me to review your manuscript. I found the paper interesting and relevant.
Introduction: Your concepts listed before your global aim page 2 line 55 did not include oxygen and phytoncides. Further the term phytoncides was not clarified as to what was actually sought in measuring them. Was it pinenes only or other substances? Please clarify here in introduction or in the methods.
Methods: Using the Delphi survey method as appropriate and detailing the use of the CVR and AHP technique enhanced the authenticity of the final tool. Adding in the oxygen, phytoncide, illuminance, UV-rays, sound and anion measurements strengthened the tool.
Results and Conclusions: Using water systems and crown density in more than one category in the tool is warranted when using the tool for forest protection and improvement. However, if using the tool as a research measure then use of concepts in more than one category can lead to multicollinearity. This should be tested with in-depth psychometrics prior to using the tool in research on forest environments for therapy with sensitive populations, like those with chronic illness.
Author Response
Thank you very much for taking the time to review this manuscript. Please find the detailed responses below and the corresponding revisions highlighted changes in the re-submitted files.

Reviewer 2 Report
Comments and Suggestions for Authors
The research presented in this article represents a significant contribution to the growing field of forest therapy. The authors have identified key factors in forest therapy and proposed specific indicators for the assessment and analysis of forest environments. Based on their analysis, they have developed an index , practical recommendations and a management plan for forest therapy, thus enriching this field of study.
However, the articulation of the article presents certain challenges that impact its readability and coherence. The current structure gives the impression that various sections, possibly written independently, have been compiled without seamless integration. This lack of cohesion makes it difficult to comprehend and would benefit from a revision to improve the transitions and connections between sections.
Regarding the methodology, there is an abrupt introduction of items and questions without proper contextualization. It would be beneficial to present these elements in advance, facilitating their understanding before they appear in the result tables. Concerning the selection of sites for field evaluation, a more robust and detailed justification is missing. It is crucial to clarify the criteria behind the choice of these four sites, including the total number of forests considered and the authority supporting their ecological value.
Additionally, the article should more clearly explain aspects such as the importance of phytoncide concentration and other tests mentioned. The connection of these elements to the study's are not incorporated in a friendly manner.
In the literature review, it would be useful to specify if the key terms were searched in the titles, abstracts, or throughout the entire text of the sources. Moreover, clarifying whether these terms were used in isolation or in combination through boolean operators is important.
The definition of an "expert" and the criteria used for selecting participants in the Delphi survey also need to be detailed. This would strengthen the validity of the selection process and the relevance of the opinions gathered.
Furthermore, it would be beneficial to detail which items and questions were evaluated by the experts and how these choices were validated. Beyond theoretical and field validation, incorporating statistical validation, such as Cronbach's Alpha, collinearity analysis, and factor analysis, among others, would be valuable to examine the interrelationship between items and reinforce the robustness of the proposed index.
Although the study offers important contributions, its impact and clarity would be enhanced by a more cohesive structure, greater transparency in its methods, and a more detailed justification of methodological choices.
Author Response

(The authors gave the same response as above.)

Reviewer 3 Report
Comments and Suggestions for Authors
Dear Author(s),
The manuscript entitled ‘‘Development of an Evaluation Index for Forest Therapy Environments’’. This is an interesting paper, and the content is very good. I took quite some time deliberating on this manuscript. I congratulate you all on the exciting manuscript I considered to be well-written. However, I have identified a few issues in the the manuscript for which I would like to call for your attention. Please find my comments and feedback below with reference to lines:
Abstract
Line 13-14, Could you please explain what you mean by four evaluation areas? When I read the next sentences, I can understand the meaning there, but the summary should still be written with clearer expressions.
Line 17- Could you please explain what the six measured fields are?
Line 23-25- It would be better if it is written that there are recommendations made.
Introduction
Line 35-46 The effect of environmental factors is mentioned here. It does not provide unity with the subject in the first paragraph. It could be related to the topic, it would be more appropriate to remove it.
Line 55-56 More information can be given about the important factors in forest therapy mentioned here. For example, why is the sunlight important? Explaining these one by one will provide basis for index development.
Line 64-65 is this opinion of the researchers or past research suggests it too? Just need a small clarification here.
Overall I think this section needs additional work and should include these key elements: (1) a discussion of previous literature with respect to forest therapy environment and evaluation index for forest therapy environments, (2) clarity in terms of what is known and what is not, and (3) statements as to what this study will present that flows from the previous two points. At the moment I don’t think there is enough background to make clear to the reader what the problem is and the importance of the study. It is hard to know what this study is building on.
Materials and Methods
I'm not very familiar with the Delphi method. However, it is not clear on what basis the indexes of the items created in Figure 1 were created. More explanation is needed here.
Line 132 General information about this method can be given below the Delphi survey.
I think it would be useful to have some research questions / hypotheses in this section in order to allow clearer interpretation of the results. The over-arching research aim is clear in the introduction, but I believe it is essential to also have some guiding research questions / hypotheses. Given this would be post-hoc, it would not be appropriate to retrofit research questions. However, I suspect the researchers had questions/hunches before data collection and these could be displayed here – this will require some thought from the researchers.
Another reason this is important is because it provides a useful baseline when looking at the results. For example, Which of the factors in forest therapy environments is more effective? How will stating and measuring this in the forest therapy evaluation index contribute to us? Having those over-arching hypotheses provide useful points to help interpret and explain the results.
Discussion
There does not appear to be a text discussing the importance of developed factors in the evaluation Index for Forest Therapy Environments. The author(s) mention these results help to provide insight into which Forest Therapy Environments are most suited for intervention.
Author Response

(The authors gave the same response as above.)

Round 2
Reviewer 2 Report
Comments and Suggestions for Authors
Thank you for submitting the revised version of your manuscript.
Just a final comment, please clearly articulate the limitations of your study (statistical validation of your index) , as well as the avenues for future research that your findings have opened. This will provide a comprehensive understanding of the scope and impact of your work.
Author Response

(The authors gave the same response as above.)
